# CD163 and pAPN double-knockout pigs are resistant to PRRSV and TGEV and exhibit decreased susceptibility to PDCoV while maintaining normal production performance

Kui Xu[1†], Yanrong Zhou[2†], Yulian Mu[1†*], Zhiguo Liu[1], Shaohua Hou[1], Yujian Xiong[2], Liurong Fang[2], Changli Ge[3], Yinghui Wei[1], Xiuling Zhang[1], Changjiang Xu[1], Jingjing Che[1], Ziyao Fan[1], Guangming Xiang[1], Jiankang Guo[1], Haitao Shang[4], Hua Li[5], Shaobo Xiao[2*], Julang Li[6*], Kui Li[1*]

[1]State Key Laboratory of Animal Nutrition and Key Laboratory of Animal Genetics, Breeding and Reproduction of Ministry of Agriculture and Rural Affairs of China, Institute of Animal Sciences, Chinese Academy of Agricultural Sciences, Beijing, China; [2]State Key Laboratory of Agricultural Microbiology and Key Laboratory of Preventive Veterinary Medicine in Hubei Province, College of Veterinary Medicine, Huazhong Agricultural University, Wuhan, China; [3]Shandong Landsee Genetics Co., Ltd., Rizhao, China; [4]Shenzhen Kingsino Technology Co., Ltd., Shenzhen, China; [5]College of Life Science and Engineering, Foshan University, Foshan, China; [6]Department of Animal BioSciences, University of Guelph, Ontario, Canada

*For correspondence:
mouyulian@caas.cn (YM);
vet@mail.hzau.edu.cn (SX);
jli@uoguelph.ca (JL);
likui@caas.cn (KL)

[†]These authors contributed equally to this work

**Abstract** Porcine reproductive and respiratory syndrome virus (PRRSV) and transmissible gastroenteritis virus (TGEV) are two highly infectious and lethal viruses causing major economic losses to pig production. Here, we report generation of double-gene-knockout (DKO) pigs harboring edited knockout alleles for known receptor proteins CD163 and pAPN and show that DKO pigs are completely resistant to genotype 2 PRRSV and TGEV. We found no differences in meat-production or reproductive-performance traits between wild-type and DKO pigs, but detected increased iron in DKO muscle. Additional infection challenge experiments showed that DKO pigs exhibited decreased susceptibility to porcine deltacoronavirus (PDCoV), thus offering unprecedented in vivo evidence of pAPN as one of PDCoV receptors. Beyond showing that multiple gene edits can be combined in a livestock animal to achieve simultaneous resistance to two major viruses, our study introduces a valuable model for investigating infection mechanisms of porcine pathogenic viruses that exploit pAPN or CD163 for entry.

## Introduction

Porcine reproductive and respiratory syndrome (PRRS) is a highly infectious viral disease characterized by reproductive disorders including premature birth, late abortion, stillbirth, weak and mummy fetuses, and respiratory dysfunction in piglets and in growing pigs (*Wensvoort et al., 1991*). Since its discovery in the United States in 1987, PRRS has rapidly spread worldwide, with frequent outbreaks causing large economic losses (*Holtkamp et al., 2013*). Three surface receptors on porcine alveolar macrophages (PAMs) have been shown to function in PRRSV invasion in vivo: heparin sulphate (HS), sialoadhesin (Sn), and CD163 (*Calvert et al., 2007*; *Crocker and Gordon, 1986*;

**eLife digest** Pig epidemics are the biggest threat to the pork industry. In 2019 alone, hundreds of billions of dollars worldwide were lost due to various pig diseases, many of them caused by viruses. The porcine reproductive and respiratory virus (PRRS virus for short), for instance, leads to reproductive disorders such as stillbirths and premature labor. Two coronaviruses – the transmissible gastroenteritis virus (or TGEV) and the porcine delta coronavirus – cause deadly diarrhea and could potentially cross over into humans. Unfortunately, there are still no safe and effective methods to prevent or control these pig illnesses, but growing disease-resistant pigs could reduce both financial and animal losses.

Traditionally, breeding pigs to have a particular trait is a slow process that can take many years. But with gene editing technology, it is possible to change or remove specific genes in a single generation of animals. When viruses infect a host, they use certain proteins on the surface of the host's cells to find their inside: the PRRS virus relies a protein called CD163, and TGEV uses pAPN.

Xu, Zhou, Mu et al. used gene editing technology to delete the genes that encode the CD163 and pAPN proteins in pigs. When the animals were infected with PRRS virus or TGEV, the non-edited pigs got sick but the gene-edited animals remained healthy. Unexpectedly, pigs without CD163 and pAPN also coped better with porcine delta coronavirus infections, suggesting that CD163 and pAPN may also help this coronavirus infect cells. Finally, the gene-edited pigs reproduced and produced meat as well as the control pigs.

These experiments show that gene editing can be a powerful technology for producing animals with desirable traits. The gene-edited pigs also provide new knowledge about how porcine viruses infect pigs, and may offer a starting point to breed disease-resistant animals on a larger scale.

*Jusa et al., 1997*). Multiple studies have reported that CD163 is an essential receptor for PRRSV infection, with scavenger receptor cysteine-rich domain 5 (SRCR5) serving as the core domain for virus recognition (*Calvert et al., 2007*; *Van Gorp et al., 2010*; *Patton et al., 2009*).

Gene editing technology has been emerging as an important approach of livestock animal and plant germplasm improvement. The technology makes possible for precise modification of more than one gene simultaneously, which is particularly desirable for obtaining important economic traits that are controlled by multiple genes. In 2016, Prather's group was the first to use CRISPR/Cas9 technology to generate SRCR5 domain-targeted *CD163* knockout pigs. They demonstrated that a *CD163* knockout line was completely resistant to genotype 2 PRRSV infection (*Whitworth et al., 2016*). Subsequently, several laboratories have generated anti-PRRSV pigs targeting *CD163*. For example, the *CD163* SRCR5 domain was replaced with human *CD163*-Like SRCR8 domain to generate PRRSV genotype 1 resistance (*Wells et al., 2017*). *Wei et al., 2018* reported homozygous gene-edited Large White pigs with a 50 bp deletion in exon 7 of the *CD163* gene (*Wei et al., 2018*) that are fully resistant to genotype 2 PRRSV. There are also examples of deletion of the SRCR5 domain seeking resistance to both PRRSV genotypes (*Burkard et al., 2017*), or introducing a premature termination in the *CD163* SRCR5 domain to generate HP-PRRSV (highly pathogenic PRRSV)-resistant Duroc pigs (*Yang et al., 2018*). Deleting the SRCR5 LBP region has also been reported to generate a PRRSV genotype 2 resistant pigs (*Guo et al., 2019*). All these studies demonstrate that PRRSV-resistant pig breeds can be generated by editing the *CD163* gene, enabling alleviation of the severity of PRRSV.

In addition to PRRSV, transmissible gastroenteritis virus (TGEV), an acute high-contact infectious virus, is known to frequently occur to co-infect with other porcine diarrhea-associated viruses such as porcine epidemic diarrhea virus (PEDV), porcine rotavirus (PoRV) (*Zhang et al., 2013*). TGEV is globally distributed and causes tremendous economic losses in pork production (*Gerdts and Zakhartchouk, 2017*). Characterized by vomiting, severe diarrhea, and dehydration, the mortality rate of TGEV-infected piglets under the age of 14 days approaches 100%. TGEV is a single-stranded, positive-sense RNA coronavirus which targets pig intestinal epithelium for infection (*Brierley et al., 1989*; *Wesley and Lager, 2003*). Studies have shown that the pAPN protein acts as a receptor in mediating TGEV infection. The viral glycoproteins bind to pAPN receptors on the surface of small intestinal epithelial cells and mediate membrane fusion, thus resulting in the virus entering into

epithelial cells (*Delmas et al., 1992*; *Hansen et al., 1998*). Inhibition or direct knockout of *pAPN* in small intestinal epithelial cells can mitigate TGEV infection (*Ji et al., 2018*; *Zhu et al., 2018*). *pAPN* knockout pigs are resistant to TGEV (*Luo et al., 2019*; *Whitworth et al., 2019*).

PDCoV is a highly virulent porcine coronavirus discovered in 2012 that causes watery diarrhea and vomiting in sows and piglets, with piglet mortality rates of 30% to 40% (*Wang et al., 2014*; *Woo et al., 2012*). There is controversy about whether or not pAPN is a functional receptor for PDCoV. *Wang et al., 2018* showed that pAPN functions as a receptor to promote PDCoV entry into cells (*Wang et al., 2018*), while *Zhu et al., 2018* confirmed its involvement but showed that pAPN was an unnecessary important functional receptor for PDCoV infection (*Zhu et al., 2018*). *Li et al., 2018* suggested that PDCoV infection may require a co-receptor, in addition to pAPN (*Li et al., 2018*). Using cells isolated from *pAPN* knockout pigs, however, *Stoian et al., 2020* showed that these pig cells were still susceptible to PDCoV infection in vitro. It was suggested that pAPN may be one of the receptors for PDCoV, and an unknown receptor or factor may compensate for pAPN function in the absence of pAPN (*Stoian et al., 2020*). However, whether *pAPN* knockout pigs may be resistant to PDCoV infection in vivo remains unknown.

Although gene-edited *CD163* knockout (PRRV resistant) pigs and *pAPN* knockout (TGEV resistant) pigs have been previously generated, respectively, pigs that are resistant to the infection of both viruses are lacking. Our objectives in the present study were (1) to knockout *CD163* and *pAPN* simultaneously using a gene editing approach; (2) to verify if the resultant DKO pigs are simultaneously resistant to infection by PRRSV and TGEV; (3) to use the DKO pigs as an in vivo experimental model to test for potential pAPN-mediated resistance to PDCoV infection. We report successfully generated gene-edited Large White pigs with both *CD163* and *pAPN* gene knockouts using CRISPR/Cas9 and somatic cell nuclear transfer (SCNT). Through viral challenge experiments, we found that these DKO pigs exhibit complete resistance to genotype 2 PRRSV and TGEV, and exhibit decreased susceptibility to PDCoV infection. In addition, with the exception of meat color score and iron content, no differences in the production performance, reproductive performance, or pork nutrient content were observed between DKO pigs and WT pigs. Thus, in addition to demonstrating that our DKO pigs are robustly resistant to both PRRSV and TGEV without suffering deleterious effects for production performance, our study also provides insights into ongoing controversy about the pAPN protein as a potential receptor for PDCoV infection of pigs.

## Results

### Generation of *CD163* and *pAPN* DKO cloned pigs

In order to generate *CD163* and *pAPN* DKO cloned pigs, we constructed sgRNA delivery plasmids targeting these genes, and selected successful DKO pig fetal fibroblasts (PEFs) as nuclear transfer donors (*Figure 1A*).

For *CD163*, the SRCR5 domain-binding site for PRRSV in exon 7 (*Van Gorp et al., 2010*; *Ma et al., 2017*) was selected as the sgRNA recognition site. To inactivate the pAPN protein, a sgRNA target site in exon two immediately downstream of the ATG start codon was selected (*Figure 1B*). Successful DKO colonies were cultured as donor cells for SCNT (*Supplementary file 1*). The cloned pigs generated in this experiment were obtained via both primary and secondary clonings. For primary cloning, the selected DKO cells are used as donors for nuclear transplantation. For secondary cloning, the ear-derived fibroblasts of the primary cloned pigs are re-cloned, which rapidly provided a large number of high-quality DKO donor cells, thus improving cloning efficiency and resulting in many genotypically identical pigs.

In our primary cloning, a total of 3780 reconstructed embryos were transplanted into 11 surrogate sows, of which two were pregnant and gave birth to eight live piglets. Of these piglets, four survived after weaning (*Figure 1C* and *Supplementary file 2*). We determined the *CD163* and *pAPN* genotypes of the four surviving piglets using PCR and Sanger sequencing. The genotypes of the three piglets (#1143, #1144, and #1145) matched that of cell colony #25, which had an 8 bp deletion on both copies of *CD163* near the target site, and a copy of *pAPN* carrying a 5 bp deletion on one copy and a 26 bp deletion on the other, both resulting in frameshift mutations or premature termination after the target site (*Figure 1D*).

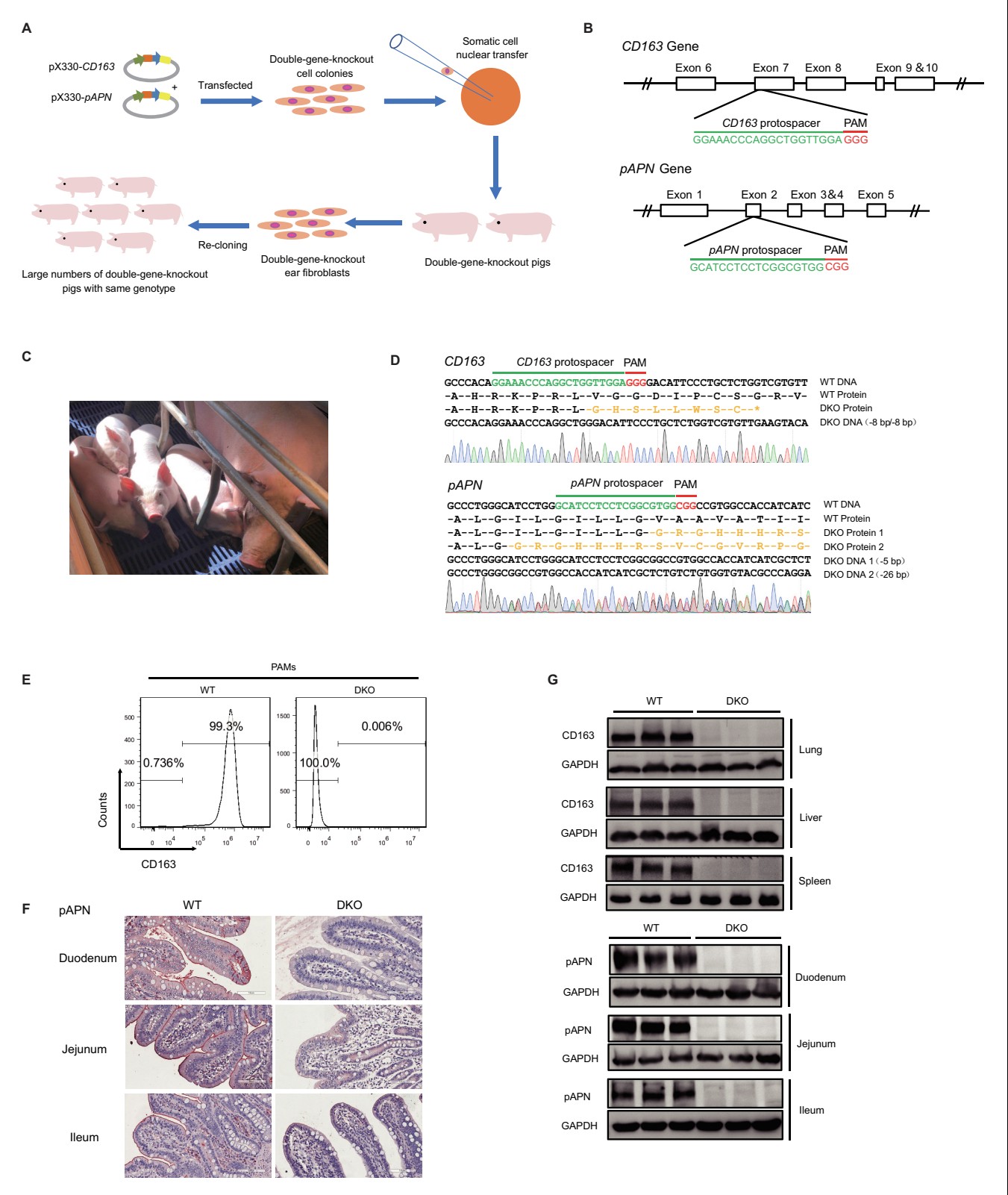

**Figure 1.** Generation of *CD163* and *pAPN* DKO pigs by CRISPR/Cas9. (**A**) Schematic overview of the generation process for DKO cloned pigs. (**B**) Genetic maps of the *CD163* (top) and *pAPN* (bottom) genes with the location and sequences of the sgRNAs. Exons, white boxes; sgRNA protospacer sequences, green; and PAM sequences, red. (**C**) Four F0 generation cloned pigs (1085#, 1143#, 1144#, 1145#) aged 1-month-old and a surrogate. (**D**) Sanger sequencing confirmation of DKO genotypes for three F0 cloned piglets (1143#, 1144#, 1145#). sgRNA protospacer sequences, green; PAM

*Figure 1 continued on next page*

*Figure 1 continued*

sequences, red; predicted amino acid sequences resulting from frameshift mutations, yellow. (E) Detection of CD163 expression on the surface of PAMs by flow cytometry. (F) Detection of pAPN expression in different small intestine segments by IHC. PAMs and tissues were derived from DKO and WT pigs. (G) Western blot analysis confirmed CD163 and pAPN expression are undetectable in different tissues of DKO pigs (N = 3).

The online version of this article includes the following figure supplement(s) for figure 1:

**Figure supplement 1.** PCR detection of random integration of the pX330 plasmid backbone.

In order to generate more DKO pigs for viral challenge experiments, we collected ear tissue samples from three piglets (#1143, #1144, and #1145) and isolated ear-derived fibroblasts. A total of 2270 reconstructed embryos generated from ear-derived fibroblasts of #1145 were transplanted into nine surrogates. Four sows successful gave birth to a total of 20 live piglets, among which 12 survived post-weaning (*Supplementary file 2*). The genotypes of these 12 piglets matched that of #1145, and the three DKO primary clones used for subsequent experiments. We used flow cytometry and western blotting for CD163, immunohistochemistry (IHC) and western blotting for pAPN, and confirmed that expression of both proteins was undetectable in DKO pigs but detectable in WT pigs of the same age and breed (*Figure 1E–G*).

We designed multiple pairs of amplification primers for the pX330 vector backbone to confirm that no random integration of pX330 vector fragments were in cloned pigs (*Figure 1—figure supplement 1*). We also tested for off-target modifications in DKO pigs using 10 potential off-target sites for each of the two sgRNAs and found no alteration in any of these 20 predicted sites in the cloned pigs (*Supplementary file 3*). This data demonstrates that clones of *Sus scrofa* line with multiple gene-edited can be generated through primary and secondary cloning with high efficiency and no off-target detected.

## DKO pigs are resistant to genotype 2 PRRSV infection

For testing of PRRSV resistance in PAMs derived from DKO pigs, we selected the highly pathogenic genotype 2 PRRSV strain WUH3 to challenge DKO and WT PAMs at a multiplicity of infection (MOI) of 0.1. qRT-PCR and western blot analyses were used to assess PRRSV proliferation in PAMs. At 12 hr post-infection (hpi), DKO PAMs carried a significantly lower PRRSV load compared with WT PAMs, and no viral RNA or PRRSV-N protein was detected thereafter in DKO PAMs (*Figure 2A and B*). The low level of PRRSV RNA that was initially detectable in the DKO line at 12 hpi is likely attributable to the adsorbed PRRSV independent of the existence of CD163, as CD163 is thought to be primarily responsible for the uncoating and viral RNA release processes of PRRSV infection (*Chen et al., 2019*; *Van Gorp et al., 2008*).

We next sought to examine if DKO pigs are resistant to PRRSV in vivo. Four 45-day-old DKO pigs and six WT control pigs of the same age were challenged with the PRRSV strain WUH3. Nasal intubation drip (2 mL: $10^6$ TCID$_{50}$/mL) and intramuscular injection (2 mL: $10^6$ TCID$_{50}$/mL) were used to infect both experimental groups. The phenotypic data of body temperature, feed intake, respiration, defecation, and mental condition were recorded daily after infection. As shown in *Figure 2C*, while fever (over 40°C) began at 1 day post-infection (dpi) and persisted throughout the remainder of the experimental period in the WT group, the body temperature of the DKO pigs stayed normal throughout the 14 days of the post-viral challenge observation period.

Scoring for other clinical symptoms of PRRSV at 1 dpi showed that WT pigs exhibited decreased appetite, shortness of breath, cough, malaise, drowsiness, and difficulty walking, whereas the DKO group displayed no abnormalities except for a brief cough and diarrhea in two pigs at 4 dpi and 9 dpi, respectively (*Figure 2D*). The body weight of the DKO pigs increased, throughout the 14 day post viral challenge observation period: the detected body weights of the WT pigs were all lower than DKO pigs after 0 dpi (*Figure 2E*). Of the six challenged WT pigs, one was slaughtered at 10 dpi to harvest PAMs, and the five remaining WT pigs died within 11 dpi. In sharp contrast, all four pigs in the DKO group remained healthy, and survived for the entire duration of the 14-day experiment (*Figure 2F*). Among the dead and slaughtered WT pigs, the lungs were swollen, with severe bleeding, and obvious lesions, while the lung tissues of dissected DKO pigs did not exhibit lesions or any other distinct symptoms associated with PRRSV (*Figure 2G*). Hematoxylin and eosin (H and E) staining showed thickening of the alveolar walls and infiltration of a large number of inflammatory

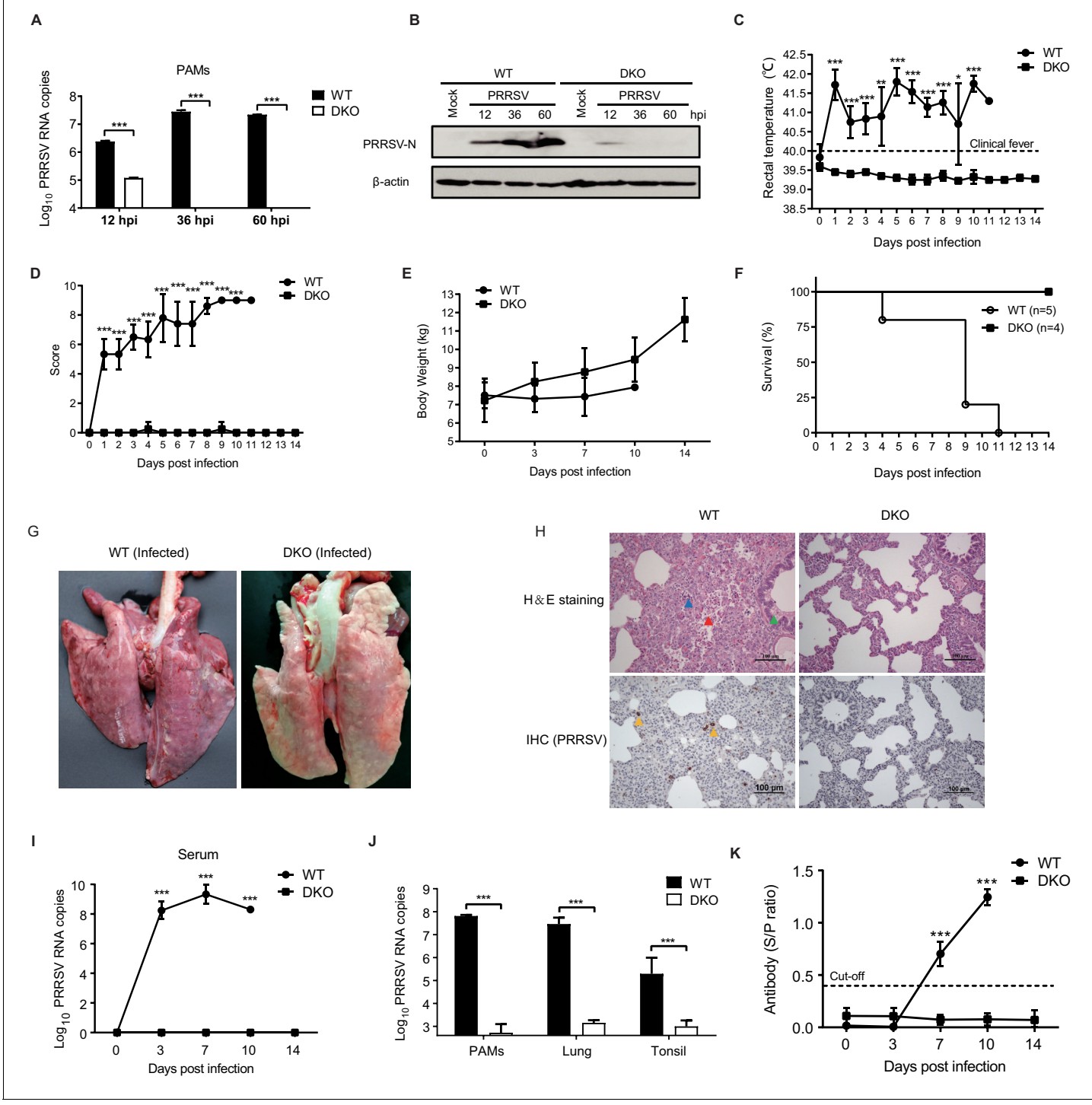

**Figure 2.** DKO pigs are resistant to genotype 2 PRRSV infection. (**A**) qRT-PCR detection of PRRSV load and (**B**) western blot confirmation of PRRSV-N protein expression in PAMs from WT and DKO pigs at 12, 36, and 60 hpi (MOI = 0.1). (**C**) Rectal temperatures and (**D**) clinical symptoms scores were recorded daily beginning at 0 dpi to 14 dpi. Temperatures > 40°C were considered fever. WT: 0 to 4 dpi, N = 6; 5 to 8 dpi, N = 5; 9 dpi, N = 4; 10 dpi, N = 2; 11 dpi, N = 1. DKO: 0 to 14 dpi, N = 4. (**E**) Body weights measured at 0, 3, 7, 10, and 14 dpi. WT: 0 to 3 dpi, N = 6; 7 dpi, N = 5; 10 dpi, N = 2. DKO: 0 to 14 dpi, N = 4. (**F**) Survival curves of WT and DKO pigs with PRRSV. WT, N = 5; DKO, N = 4. (**G**) Representative lesions of infected lungs. (**H**) H and E staining (top) of lesions in lung tissue; IHC (bottom) detection of viral antigens in PRRSV-infected lungs. Lymphocytic infiltration (blue triangle); necrotic cells in the alveolar wall (red triangle); bronchial wall dilated and filled with serous fluid (green triangle); PRRSV-N protein (yellow triangle) (**I**) PRRSV loads in serum at 0, 3, 7, 10, and 14 dpi. WT: 0 to 3 dpi, N = 6; 7 dpi, N = 5; 10 dpi, N = 2. DKO: 0 to 14 dpi, N = 4. (**J**) PRRSV loads in PAMs (WT group: N = 3; DKO group: N = 3), lung tissues (WT: N = 5; DKO: N = 3), and tonsil tissues (WT: N = 5; DKO: N = 4). (**K**) PRRSV-specific antibodies

*Figure 2 continued on next page*

*Figure 2 continued*

in serum. WT: 0 to 3 dpi, N = 6; 7 dpi, N = 5; 10 dpi, N = 2. DKO: 0 to 14 dpi, N = 4. Data are expressed as means ± SD. Statistical significance was determined by Student's *t* test; ns, p>0.05; *p<0.05; **p<0.01; ***p<0.001.

The online version of this article includes the following source data for figure 2:

**Source data 1.** The qRT-PCR detection of PRRSV load in PAMs.
**Source data 2.** Rectal temperatures of pigs.
**Source data 3.** Clinical symptoms scores of pigs.
**Source data 4.** Body weights of pigs.
**Source data 5.** The survival rate of pigs.
**Source data 6.** The qRT-PCR detection of PRRSV load in serum.
**Source data 7.** The qRT-PCR detection of PRRSV load in PAMs, lung tissues and tonsil tissues.
**Source data 8.** PRRSV-specific antibodies in serum (S/P ration).

cells in the pulmonary interstitium of the WT pig lungs, while no pathological changes were found in the lung tissue of DKO pigs (*Figure 2H*, upper panel).

Examination of PRRSV antigens in lung tissue via IHC, it was revealed that the viral antigens were present in the lungs of the WT group, but not that of the DKO pigs (*Figure 2H*, lower panel). Moreover, we measured the PRRSV viral load in the serum of both groups at 0, 3, 7, 10, and 14 dpi and found that in the WT group, the PRRSV load increased rapidly and significantly by 7 dpi, reaching its maximum at 7 dpi. In agreement with other experiments showing viral resistance, the PRRSV viremia in the DKO group remained negative throughout the challenge (*Figure 2I*). We also tested the PRRSV viral load in PAMs, lung tissues, and tonsil tissues of the two groups of pigs after viral challenge. Whereas a high titer of PRRSV was detected in all tissues examined in the WT group, PRRSV was almost undetectable in DKO pigs (*Figure 2J*). From 3 dpi, the amount of PRRSV-specific ELISA antibodies in the serum of WT pigs increased significantly, and antibody levels were positive (S/P≥0.4) at 7 and 10 dpi, while such antibodies in DKO pigs remained consistently negative (S/P<0.4) (*Figure 2K*). Taken together, these results provide compelling in vitro and in vivo evidence that the DKO pigs are resistant to PRRSV infection.

## DKO pigs are resistant to TGEV infection

Following characterization of PRRSV resistance, we next sought to determine if double knockout of *CD163* and *pAPN* also conferred resistance against TGEV. Four 45-day-old DKO pigs and six WT control pigs of the same age and breed were fed under the same conditions and infected with TGEV. A total of 10 mL of TGEV ($7 \times 10^5$ TCID$_{50}$/mL) were orally administered to each pig in two doses (day 0 and day 1, 5 mL/day). At 3 dpi, one DKO pig and one WT pig were slaughtered to collect intestinal tissues for pathological examination, and the remaining pigs were housed under regular husbandry conditions until slaughter, and tissues were sampled at 14 dpi. Body temperature was recorded daily beginning at Day 0, prior to inoculation, and piglet weighing and blood sampling for serum separation were conducted at 0, 7, and 14 dpi.

During the viral challenge period, no abnormalities were observed among the pigs, with the exception of two WT pigs that had diarrhea. There was no significant difference in weight gain between the two groups (data not shown). Detection of TGEV-specific neutralizing antibodies in serum showed no neutralizing antibodies in the DKO pigs throughout the experiment, while two of the WT pigs were positive for neutralizing antibodies at 7 dpi, and all WT pigs were positive by 14 dpi (*Figure 3A*).

All slaughtered pigs from both WT and DKO groups (sampled at 3 dpi and 14 dpi) were dissected to examine potential lesions in small intestine tissues. For the DKO group, no lesions were found in the small intestine samples collected at either 3 dpi or 14 dpi (*Figure 3B*). In marked contrast, WT group tissues collected at 3 dpi demonstrated a thin and yellowing small intestine wall, with hemorrhages typical of TGEV clinical symptoms, and by 14 dpi there were notable duodenum, jejunum, and ileum hemorrhages, accompanied by intestinal wall thinning and enlarged mesenteric lymph nodes (*Figure 3B*). Pathological examination of small intestine tissue sections revealed pathological changes, including necrosis and shedding of intestinal mucosal epithelial cells, intestinal villi fusion, plasma cells accumulating in the lamina propria, and infiltration of eosinophils in the duodenum,

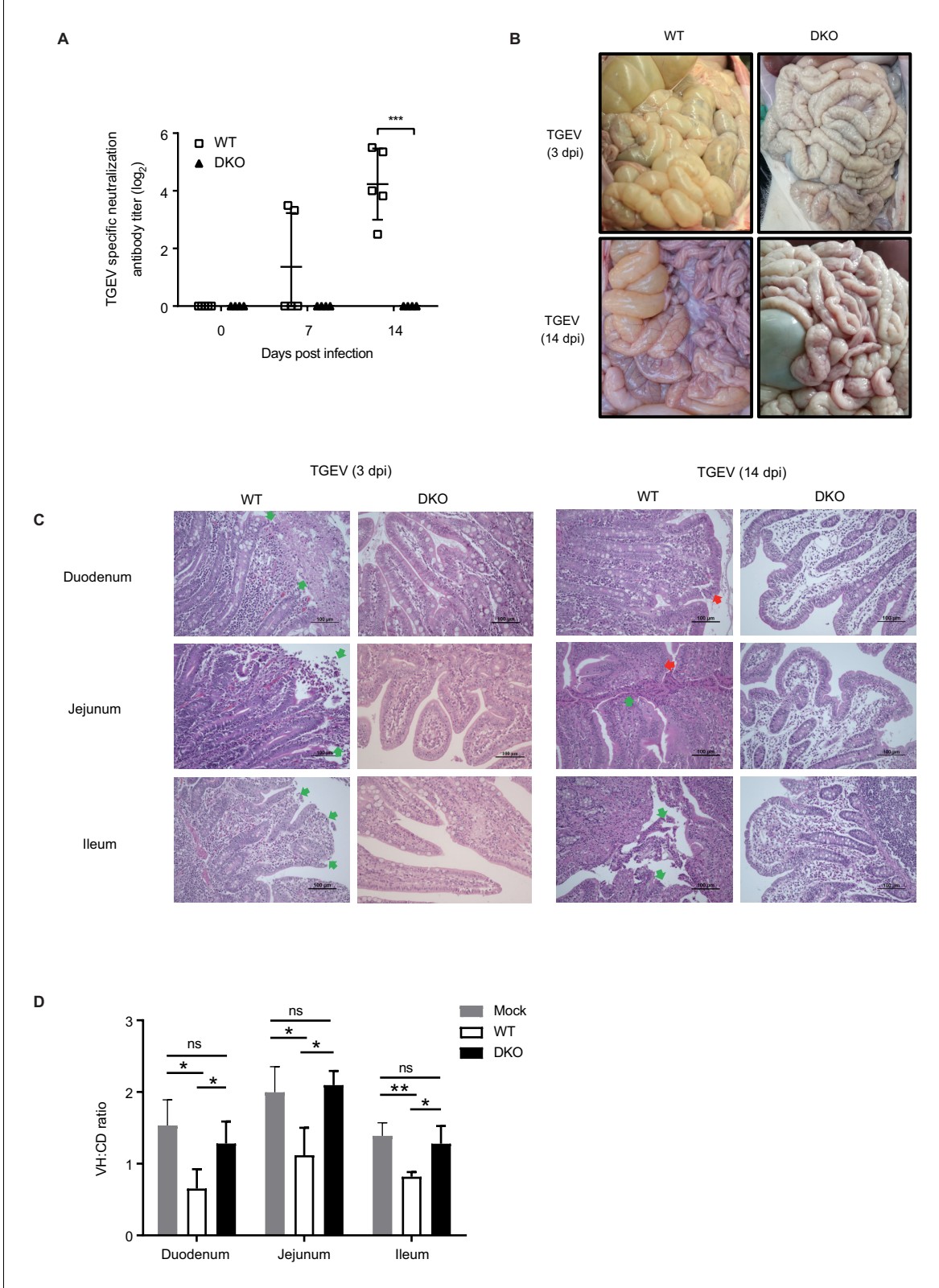

**Figure 3.** DKO pigs are resistant to TGEV infection. (**A**) TGEV-specific antibody detection in serum of WT and DKO pigs at 0, 7, and 14 dpi. WT group: N = 5; DKO group: N = 4. (**B**) Representative macroscopic lesion of small intestines at 3 dpi (top) and 14 dpi (bottom) in WT and DKO pigs. (**C**) H and E staining of lesions small intestine sections derived from both WT and DKO pigs at 3 dpi (left) and 14 dpi (right). H and E staining shows intestinal villi fusion, plasma cells accumulating in the lamina propria, and infiltration of eosinophils (red arrow), and necrosis and shedding of intestinal mucosal

*Figure 3 continued on next page*

*Figure 3 continued*

epithelial cells (green arrow). (D) The ratio of intestinal villus height to the crypt depth derived from both WT and DKO pigs at 3 dpi. Mock: duodenum to ileum, N = 3. WT: duodenum to ileum, N = 3. DKO: duodenum, N = 4; jejunum, N = 3; ileum, N = 4. Data are expressed as the mean ± SD. Statistical significance was determined by Student's *t*-test; ns, p>0.05; *p<0.05; **p<0.01; ***p<0.001.

The online version of this article includes the following source data for figure 3:

**Source data 1.** TGEV-specific neutralization antibody in serum.
**Source data 2.** The ratio of intestinal villus height to the crypt depth in TGEV group.

jejunum, and ileum of WT pigs at 3 dpi and 14 dpi, while the same small intestine tissues in DKO pigs appeared healthy (*Figure 3C*). We also analyzed the ratio of intestinal villus height (VH) to the crypt depth (CD). The smaller the ratio, the more severe the intestinal villi atrophy. We found that compared with the mock group, the three intestinal segments of the WT group had significant intestinal villous atrophy, and the intestinal villi of these intestinal segments in the DKO group did not show atrophy; that is, the degree of intestinal villous atrophy in the three intestine segments in the WT group was significantly higher than that in the DKO group (*Figure 3D*). These results consistently demonstrate that our *CD163/pAPN* DKO pigs exhibit strong resistance to TGEV infection.

## DKO pigs exhibit decreased susceptibility to PDCoV infection

PDCoV is a highly pathogenic virus that has recently been shown to cause diarrhea in newborn piglets, although the functional receptors for PDCoV have not yet been confirmed (*Li et al., 2018*; *Stoian et al., 2020*; *Zhu et al., 2018*). Whether pAPN functions as a receptor or co-receptor in PDCoV infection of pigs remains controversial. To test the hypothesis that pAPN may functionally mediate PDCoV infection, we tested the susceptibility of our DKO pigs to this virus. Two 45-day-old DKO pigs and four WT pigs of the same age and breed were challenged with PDCoV. A total of 16 mL of PDCoV ($2.5 \times 10^8$ TCID$_{50}$/mL) was orally administered to each pig in two doses (Day 0 and Day 1, 8 mL/day).

During the 14 days of PDCoV challenge study, both the DKO and WT pigs appeared normal, with no distinct differences in body temperature or weight (data not shown). Blood was collected at 0, 7, and 14 dpi to assay for levels of virus-specific antibodies. At 7 and 14 dpi, WT pigs were all antibody-positive, while the DKO pigs were all antibody-negative at 7 dpi, but carried antibody levels comparable to that of the WT group by 14 dpi (*Figure 4A*). This suggests that the double-gene knockout led to a delayed onset of humoral immunity in pigs, possibly due to delayed-immune system exposure to the virus. All pigs were slaughtered at 14 dpi, and the small intestine tissues were collected to evaluate disease severity. It was found that the intestinal wall of the WT had become thinner, with watery fluid in the small intestine, and mesenteric hyperemia, none of which was observed in the small intestine of the DKO pigs (*Figure 4B*).

Pathological examination of small intestine tissue sections revealed significant lesions in the small intestine tissues of both of the WT and DKO groups, which included intestinal villi fusion, infiltration of lymphocytes in the intestinal mucosa, with many lesions in the intrinsic membrane in the duodenum and jejunum tissues. In the ileum, there were signs of necrosis and shedding of intestinal mucosal intraepithelial cells and naked lamina propria. The extent of lesions in the WT pigs was more severe than that of the DKO pigs (*Figure 4C*). We also detected the ratio of intestinal villus height to the crypt depth, and found that compared with the mock group, the three intestinal segments of both of the WT group and the DKO group had intestinal villous atrophy, but the degree of villous atrophy in the ileal tissue in the DKO group was lower than that of the WT group (*Figure 4D*). In addition, we tested the resistance of PAMs derived from DKO pigs to PDCoV. DKO and WT PAMs were infected with PDCoV, and indirect immunofluorescence assays (IFA), tissue culture infectious dose 50 (TCID$_{50}$) assays, qRT-PCR, and western blot analyses to assess PDCoV proliferation in PAMs all indicated that DKO PAMs exhibit significantly decreased susceptibility of PDCoV infection compared to WT PAMs (*Figure 4—figure supplement 1*). These data suggest that although the DKO line is still susceptible to PDCoV infection, the viral invasion and damage to the small intestines was partially inhibited compared to that of the WT line.

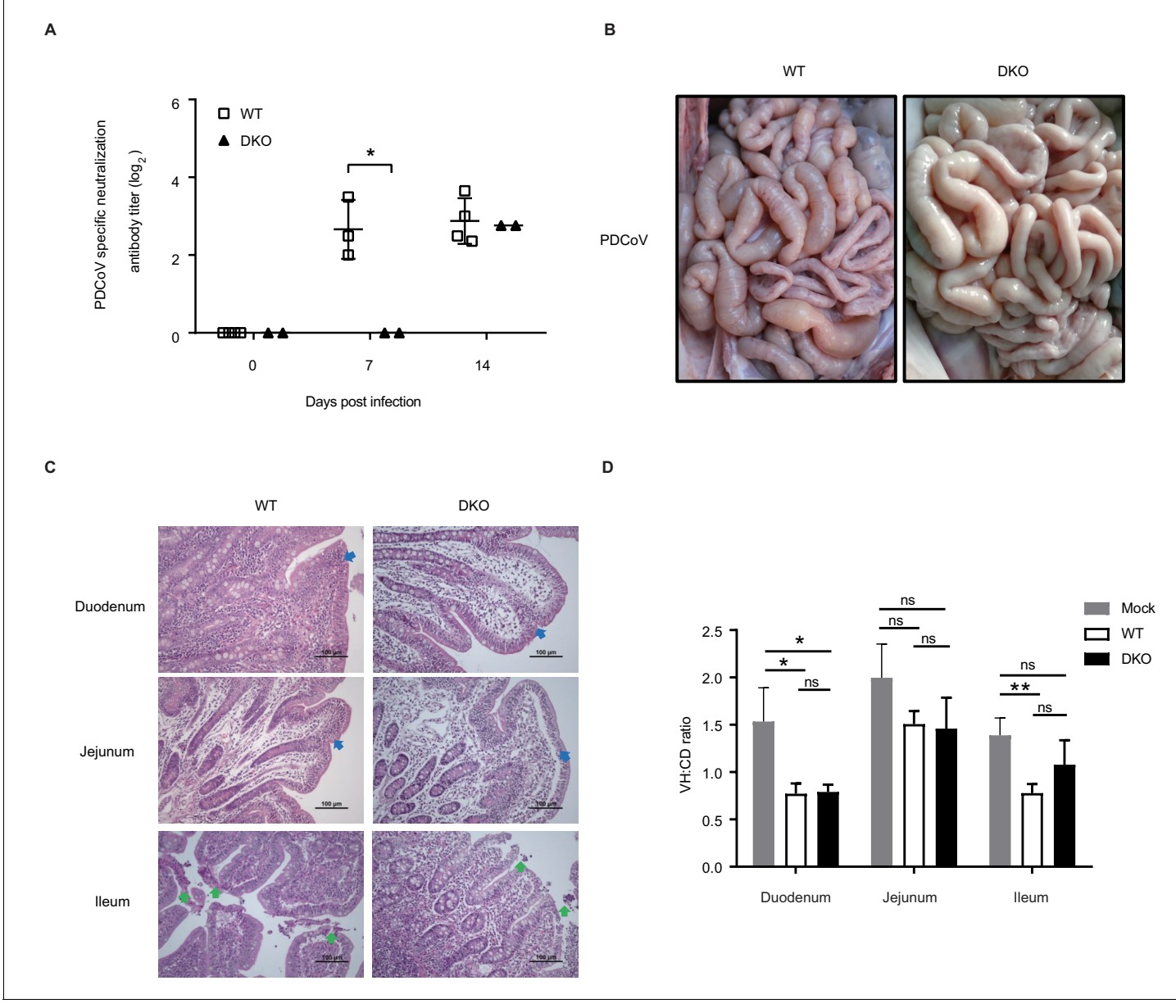

**Figure 4.** DKO pigs exhibit reduced susceptibility to PDCoV. (A) PDCoV-specific antibody detection in serum from both WT and DKO pigs at 0, 7, and 14 dpi. WT group: 0 dpi, N = 4; 7 dpi, N = 3; 14 dpi, N = 4. DKO group, N = 2. (B) Representative macroscopic lesions of the small intestines at 14 dpi from WT and DKO groups of pigs challenged with PDCoV. (C) H and E staining of small intestine segments sections to detect lesions; small intestine tissues were derived from PDCoV-infected WT and DKO pigs. Intestinal villi fusion, infiltration of lymphocytes in the intestinal mucosa, and a large number in the intrinsic membrane (blue arrow), and necrosis and shedding of intestinal mucosal intraepithelial cells and naked lamina propria (green arrow). (D) The ratio of intestinal villus height to the crypt depth derived from both WT and DKO pigs at 14 dpi. Mock: duodenum to ileum, N = 3. WT: duodenum to ileum, N = 3. DKO: duodenum to ileum, N = 3. Data are expressed as the mean ± SD. Statistical significance was determined by Student's *t*-test; ns, p>0.05; *p<0.05; **p<0.01; ***p<0.001.

The online version of this article includes the following source data and figure supplement(s) for figure 4:

**Source data 1.** PDCoV-specific neutralization antibody in serum.
**Source data 2.** The ratio of intestinal villus height to the crypt depth in PDCoV group.
**Figure supplement 1.** PAMs of DKO pigs exhibit reduced susceptibility to PDCoV.
**Figure supplement 1—source data 1.** The TCID$_{50}$ detection of PDCoV titer in PAMs.
**Figure supplement 1—source data 2.** The qRT-PCR detection of PDCoV RNA copies in PAMs.
**Figure supplement 1—source data 3.** The western blot detection of the relative PDCoV-N protein level in PAMs.

**Table 1.** Carcass traits and meat quality characteristics of 11-month-old DKO and WT Large White pigs.

| Item | Mean ± SEM of WT | Mean ± SEM of DKO | p Value | |
|---|---|---|---|---|
| Live weight at slaughter (kg) | 160.6 ± 7.371 N=3 | 163.6 ± 3.215 N=3 | 0.7280 | ns |
| Carcass weight (kg) | 122.3 ± 6.930 N=3 | 127.2 ± 2.242 N=3 | 0.5433 | ns |
| Carcass length (cm) | 118.6 ± 1.468 N=3 | 117.6 ± 0.8988 N=3 | 0.6164 | ns |
| Dressing percentage (%) | 76.11 ± 0.9685 N=3 | 77.74 ± 0.2221 N=3 | 0.1763 | ns |
| Ham percentage (%) | 31.66 ± 0.6855 N=3 | 31.74 ± 0.5382 N=3 | 0.9313 | ns |
| Lean rate (%) | 69.94 ± 0.9530 N=3 | 66.74 ± 0.9995 N=3 | 0.08160 | ns |
| Loin eye area (cm$^2$) | 55.79 ± 3.145 N=3 | 61.72 ± 3.515 N=3 | 0.2766 | ns |
| Average backfat thickness (mm) | 15.22 ± 0.4129 N=3 | 17.97 ± 2.076 N=3 | 0.2628 | ns |
| Muscle pH 1 | 6.100 ± 0.07211 N=3 | 6.183 ± 0.1676 N=3 | 0.6715 | ns |
| Muscle pH 24 | 5.673 ± 0.01764 N=3 | 5.670 ± 0.01000 N=3 | 0.8774 | ns |
| Meat color score | 3.833 ± 0.1667 N=3 | 4.667 ± 0.1667 N=3 | 0.02410 | * |
| Marbling | 1.167 ± 0.1667 N=3 | 1.167 ± 0.1667 N=3 | >0.9999 | ns |
| Drip loss (%) | 3.547 ± 0.3310 N=3 | 3.257 ± 0.1690 N=3 | 0.4788 | ns |

ns, p>0.05; *p<0.05.

The online version of this article includes the following source data for Table 1:

Source data 1. Carcass traits and meat quality characteristics of DKO and WT Large White pigs.

## DKO pigs maintain normal production performance

We next evaluated the growth and performance indices of DKO pigs. Three 11-month-old DKO Large White boars and three WT Large White boars of the same age were selected for slaughter testing. The live weight at slaughter, carcass weight and length, dressing percentage, ham percentage, lean rate, loin eye area, average backfat thickness, muscle pH, marbling, and drip loss were determined. As shown in *Table 1*, with the exception of meat coloring score, DKO pigs showed no difference in comparison with WT pigs for these indices. In addition, there was no significant difference in birth weight or in the average daily gain between WT and DKO pigs (*Supplementary file 8*). Most notably, the meat color score in the DKO pigs (4.667 ± 0.1667 N=3) was significantly higher than that of WT pigs (3.833 ± 0.1667 N=3), although both were within the normal range of 2 to 5 according to the guideline of 'rules for performance testing of breeding pigs' document published by the Ministry of Agriculture and Rural Affairs of PR China (NY/T 821–2004) (*Table 1* and *Figure 5A–B*). Since the CD163 protein is known to play a role in the degradation of haemoglobin-haptoglobin (Hb-Hp), and considering that Fe is an important component of haemoglobin, we reasoned that the increased meat color score (redness) may be due to the decreased Hb metabolism as a consequence of *CD163* knockout, and subsequently mild accumulation of Fe containing Hb in the meat.

To test this hypothesis, the meat Fe level was analyzed, it was found that the concentration of Fe was significantly higher in DKO pigs compared to WT pigs (*Figure 5C*). We also tested the serum haptoglobin (Hp) content and found that the Hp content in DKO pigs was significantly higher than that of WT control pigs (*Figure 5D*). Evaluation of the nutritional components of pork such as total protein, total fat, ash, moisture, specific minerals, and amino acid content was also performed. As shown in *Table 2* and *Supplementary file 4*, no differences in these indices were observed between the two groups.

In order to test the reproductive performance of the DKO boars, semen from DKO male pigs (n = 3) and that of WT pigs (n = 4) of the same age and breed were analyzed. It was revealed that the concentration, motility, and velocity distribution of the sperm from DKO boars did not differ from WT boars (*Table 3*). Furthermore, there was no difference in the litter size between the two genotypes: DKO litters were 10.67 ± 1.202 (N = 3, litter size from 9 to 13) and the WT litters were

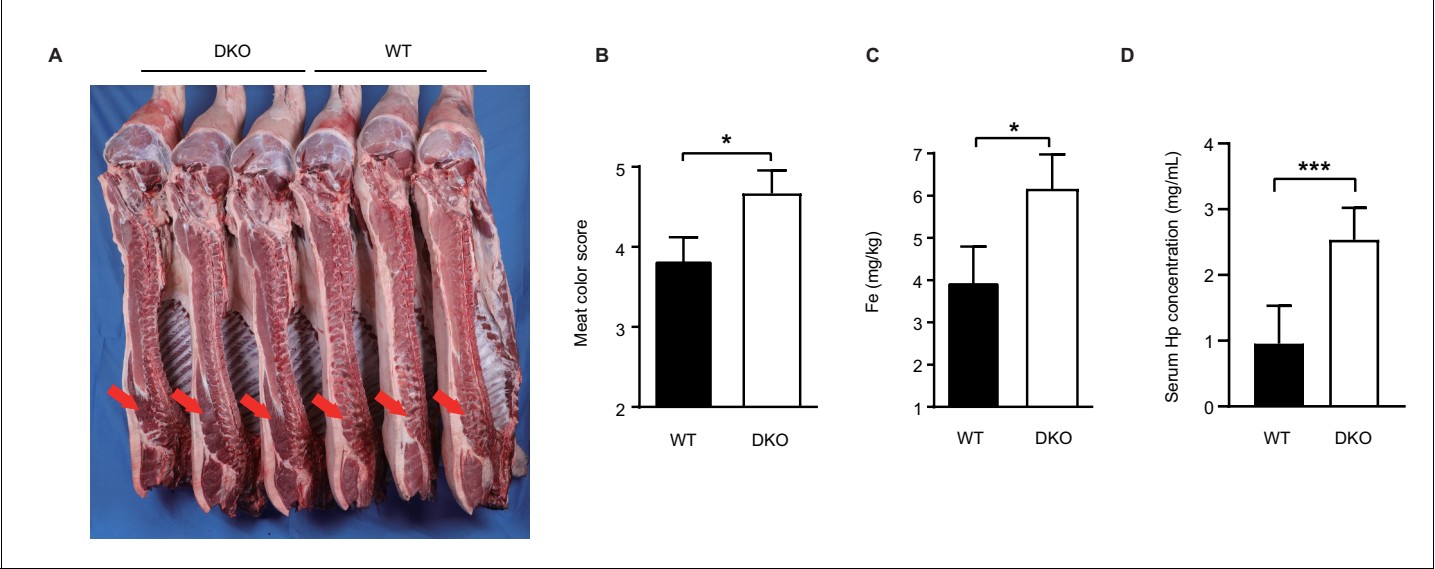

**Figure 5.** DKO pigs maintain normal production performance. (**A**) Carcass photo of DKO and WT pigs. The meat of DKO pigs becomes darker red (red arrow). (**B**) Meat color scores for WT and DKO pigs. WT group: N = 3; DKO group: N = 3. (**C**) Detection of Fe content in longissimus dorsi muscle from both experimental groups. WT group: N = 3; DKO group: N = 3. (**D**) Serum Hp levels of WT and DKO pigs. WT group: N = 22; DKO group: N = 6. Data are expressed as the mean ± SD. Statistical significance was determined by Student's t-test; ns, p>0.05; *p<0.05; **p<0.01; ***p<0.001. The online version of this article includes the following source data for figure 5:

**Source data 1.** Meat color scores for WT and DKO pigs.
**Source data 2.** Detection of Fe content in longissimus dorsi muscle from WT pigs and DKO pigs.
**Source data 3.** Serum Hp levels of WT and DKO pigs.

12.05 ± 0.6496 (N = 22, litter size from 7 to 17). In addition, these three DKO pigs did not show any growth abnormalities or disease phenomena during the 11-month rearing process, and no abnormalities were observed in the main tissues and organs after slaughter (data not shown). Taken together, with the exception of slight meat coloring score increase, these results show that the simultaneous, editing-based disruption of the *CD163* and *pAPN* loci, does not affect the normal growth and reproductive performance of the resultant DKO pigs.

## Discussion

Conventional breeding for complex traits using molecular marker-assisted selection is a lengthy process, requiring multiple rounds of crosses and backcrosses to introgress each individual gene. CRISPR/Cas9 gene editing not only allows bypassing of this long process, but also provides a possible means to obtain multiple beneficial genotypes in a single generation while also avoiding gene penetration from donor species, thus maintaining the desirable qualities of the original species.

*Zhou et al., 2015* first used CRISPR/Cas9 in combination with SCNT to generate knockout of *PARK2* and *PINK1* genes, whose dysfunction are known to contribute to the early onset of Parkinson's disease in humans (*Zhou et al., 2015*). *Huang et al., 2017* got the pig model with metabolic disorder successfully by editing apolipoprotein E and low density lipoprotein receptor genes simultaneously (*Huang et al., 2017*). Our study is the first report on how multiple gene edits can be combined in livestock animal to offer simultaneous resistance to two major viral infection. Similar to the previous reports above, double knockout efficiency using CRISPR/Cas9-mediated dual gene editing method without any drug or flow cytometry screening was high in our study, reaching 6.30% (17 DKO cell colonies out of 270 cell colonies). In this experiment, we quickly generated a large number of DKO pigs by re-cloning. We found that the re-cloning efficiency (0.9%, 20/2270) was much higher than the primary cloning efficiency (0.2%, 8/3780). A possible reason for this elevated efficiency could be that the monoclonal cells used for the primary cloning must be cultured in vitro for a long

**Table 2.** Total protein, total fat, ash, moisture, and individual mineral content of DKO lean meat and WT lean meat.

| Item | Mean ± SEM of WT | Mean ± SEM of DKO | p Value | |
|------|------------------|-------------------|---------|---|
| Total protein (%) | 23.40 ± 0.2517 N=3 | 23.03 ± 0.3930 N=3 | 0.4760 | ns |
| Total fat (%) | 1.100 ± 0.1528 N=3 | 2.033 ± 0.3712 N=3 | 0.0807 | ns |
| Ash (%) | 1.100 ± 0 N=3 | 1.100 ± 0 N=3 | | |
| Moisture (%) | 73.73 ± 0.6360 N=3 | 72.90 ± 0.6083 N=3 | 0.3973 | ns |
| P (mg/100 g) | 220.0 ± 2.000 N=3 | 219.3 ± 3.333 N=3 | 0.8722 | ns |
| Ca (mg/kg) | 36.33 ± 2.134 N=3 | 43.17 ± 6.598 N=3 | 0.3802 | ns |
| Cu (mg/kg) | <0.5000 | <0.5000 | | |
| Fe (mg/kg) | 3.953 ± 0.4872 N=3 | 6.160 ± 0.4701 N=3 | 0.0311 | * |
| K (g/kg) | 3.293 ± 0.2972 N=3 | 2.897 ± 0.1821 N=3 | 0.3186 | ns |
| Mg (mg/kg) | 260.3 ± 0.3333 N=3 | 262.3 ± 1.202 N=3 | 0.1841 | ns |
| Mn (mg/kg) | <0.3000 | <0.3000 | | |
| Na (mg/kg) | 353.3 ± 6.360 N=3 | 356.0 ± 14.50 N=3 | 0.8744 | ns |
| Se (mg/kg) | 0.1013 ± 0.004667 N=3 | 0.08200 ± 0.008660 N=3 | 0.1208 | ns |

ns, $p > 0.05$; *$p < 0.05$.

The online version of this article includes the following source data for Table 2:

Source data 1. Total protein, total fat, ash, moisture, and individual mineral content of DKO lean meat and WT lean meat.

time, which has been reported to inhibit cloning efficiency (*Li et al., 2003*; *Magnani et al., 2008*; *Mastromonaco et al., 2006*). The donor cells used in re-cloning were ear-derived fibroblasts isolated directly from DKO pigs, eliminating the requirement for a long-term, in vitro screening process. Our findings support the notion that the efficiency of this approach is not gene specific, and may be applicable to the knockout of other genes that allow improving disease resistance or animal production.

In 2007, Calvert et al. first discovered that CD163 functions as a PRRSV receptor protein during PAMs infection, which has since been confirmed by several studies (*Calvert et al., 2007*; *Van Gorp et al., 2008*; *Guo et al., 2014*; *Patton et al., 2009*). Structural studies of CD163 revealed that the SRCR5 domain corresponding to *CD163* exon seven is necessary to mediate PRRSV infection (*Van Gorp et al., 2010*). In recent years, several groups have successfully generated PRRSV-resistant gene-edited pigs by targeting exon 7 of the pig *CD163* gene (*Burkard et al., 2017*; *Guo et al.,*

**Table 3.** Comparison of the concentration, motility, and velocity distribution of the sperm between DKO and WT Large White pigs.

| Item | Mean ± SEM of WT | Mean ± SEM of DKO | p Value | |
|------|------------------|-------------------|---------|---|
| Concentration ($10^6$/mL) | 1089 ± 137.9 N=4 | 1176 ± 231.5 N=3 | 0.7461 | ns |
| Motility (%) | 88.00 ± 2.345 N=4 | 86.00 ± 3.215 N=3 | 0.6268 | ns |
| Rapid (%) | 2.000 ± 1.683 N=4 | 2.500 ± 0.7638 N=3 | 0.8206 | ns |
| Medium (%) | 33.25 ± 1.702 N=4 | 35.50 ± 11.79 N=3 | 0.8320 | ns |
| Slow (%) | 52.25 ± 2.926 N=4 | 48.00 ± 9.504 N=3 | 0.6449 | ns |
| Static (%) | 12.00 ± 2.345 N=4 | 14.00 ± 3.215 N=3 | 0.6268 | ns |

ns, $p > 0.05$.

The online version of this article includes the following source data for Table 3:

Source data 1. Comparison of the concentration, motility, and velocity distribution of the sperm between DKO and WT Large White pigs.

*2019*; *Whitworth et al., 2016*; *Yang et al., 2018*). In the present study, we used a single sgRNA targeting exon 7 of *CD163*, generated an 8 bp double-stranded deletion that terminated protein translation near the target site. Our finding on the complete resistance to PRRSV genotype 2 in our knockout line is consistent with those previous reports.

CD163 is known to play a role in promoting the clearance of plasma free haemoglobin (*Kristiansen et al., 2001*). Our finding that the DKO pigs have higher meat Fe content and have elevated serum Hp levels is consistent with this idea, and may explain the observed darker red color in our DKO meat. Interestingly, and consistent to our finding, *Wells et al., 2017* also reported that the serum Hp levels are elevated in *CD163* knockout pigs (*Wells et al., 2017*). Despite the slight color score increase, no abnormal growth or reproductive performance was observed in our DKO pigs, and the meat color of both DKO pigs and WT pigs were within the normal range. Production performance evaluations and identification of pork nutritional components showed that our DKO pigs were indistinguishable from that of the WT pigs in growth rate and reproductive performances, except for the meat color score and iron content. However, the number of DKO pigs tested by us is still small, and the production performance of DKO pigs still needs to be verified in large populations in the future.

APN is known to be a receptor for many coronaviruses, and studies have shown that separate domains function in virus recognition vs. hydrolase catalytic activity (*Reguera et al., 2012*). Two research groups have recently demonstrated that *pAPN* knockout pigs block TGEV but not PEDV infection (*Luo et al., 2019*; *Whitworth et al., 2019*). Our data showing that *pAPN* knockout can completely prevent TGEV virus infection are consistent with these recently published findings. In addition to TGEV and PRRSV, we also determined if *pAPN* deletion conferred protection against PDCoV. APN is a receptor for multiple coronaviruses and is abundantly expressed on small intestinal epithelial cells, which has led to the speculation that pAPN may also be a receptor for PDCoV. *Wang et al., 2018* and *Li et al., 2018* proposed that pAPN functions as a receptor in mediating PDCoV infection (*Li et al., 2018*; *Wang et al., 2018*). However, another study found that knockout of *pAPN* in IPI-2I cells inhibited but did not completely block PDCoV infection, suggesting that pAPN was not essential for viral recognition (*Zhu et al., 2018*). Taken together, these studies suggest that pAPN may be involved in PDCoV infection, but PDCoV may also be able to enter cell through other pathway(s).

Our results on the delayed PDCoV-specific neutralizing antibodies production, and a reduced extent of gross and histopathological lesions on small intestine in DKO pigs compared to WT pigs are consistent with this previous suggestion that pAPN may play a role but is not the only path for PDCoV cell entry. Interestingly, a recent study showed that PAMs, but not lung fibroblast-like cells, from *pAPN* knockout pigs showed resistance to PDCoV infection (*Stoian et al., 2020*), a finding consistent with our in vitro experiments showing that DKO PAMs exhibit decreased susceptibility to PDCoV infection. In addition, *pAPN* knockout pigs are susceptible to PDCoV when virus levels were detected using qRT-PCR, and virus neutralization activity was measured, although the extent of tissue lesions between the KO and WT groups was not compared (*Stoian et al., 2020*). Our findings are in line with this study reporting that *pAPN* knockout pigs are still susceptible to PDCoV. However, as reflected by the delay in neutralizing antibody response, and much lighter intestine damage in the DKO pigs, the susceptibility of the *pAPN* knockout group to the virus is reduced compared that of the WT pigs, indicating the potential role of pAPN in mediating PDCoV infection. Additionally, the effect of *CD163* knockout in the delayed adaptive immune response cannot be ignored. Despite the important role of CD163 in innate immunity, an inhibiting effect of soluble CD163 on the adaptive immune system has also been reported (*Frings et al., 2002*; *O'Connell et al., 2017*). It is thus possible that the delayed adaptive immune response we observed in PDCoV-infected DKO pigs may be associated with *CD163* knockout-induced immunosuppression.

In summary, the DKO pigs generated in this study are simultaneously resistant to PRRSV and TEGV, and exhibit decreased susceptibility to PDCoV, while maintaining the same growth and reproductive production traits when compared to WT animals. These pigs may offer breeding starting points for disease-resistant pig colony generation and will be a valuable model to help deepen our understanding of the role and mechanisms of these receptor proteins in the infection mechanisms of multiple viruses.

# Materials and methods

## Key resources table

| Reagent type (species) or resource | Designation | Source or reference | Identifiers | Additional information |
|---|---|---|---|---|
| Gene (*Sus scrofa*) | *CD163* | *Sus scrofa* (pig) Genome Database | GenBank: NC_010447 | |
| Gene (*Sus scrofa*) | *pAPN* | *Sus scrofa* (pig) Genome Database | GenBank: NC_010449.5 | |
| Strain, strain background (highly pathogenic porcine reproductive and respiratory syndrome virus (HP-RRSV)) | WUH3 | *Li et al., 2009* | GenBank: HM853673 | |
| Strain, strain background (transmissible gastroenteritis virus (TGEV)) | WH-1 | *An et al., 2014* | GenBank: HQ462571 | |
| Strain, strain background (Porcine deltacoronavirus (PDCoV)) | CHN-HN-2014 | *Dong et al., 2016* | GenBank: KT336560 | |
| Genetic reagent (*Sus scrofa*) | Large White pigs | Ninghe national original pig farm | | The Large White pigs used in this experiment were of the same strain |
| Cell line (*Sus scrofa*) | pig fetal fibroblasts (PEFs) (Large White pigs) | *Ruan et al., 2015* | | Primary cell line: PEFs were isolated from 35-day-old male Large White pigs |
| Cell line (*Sus scrofa*) | porcine alveolar macrophages (PAMs) (Large White pigs) | *Wensvoort et al., 1991* | | Primary cell line: PAMs were isolated from DKO Large White piglets and WT Large White piglets |
| Cell line (*Sus scrofa*) | ST cells | ATCC | ATCC: CRL-1746; RRID:CVCL_2204 | |
| Cell line (*Sus scrofa*) | LLC-PK1 cells | ATCC | ATCC: CL-101; RRID:CVCL_0391 | |
| Antibody | anti-CD163 (rabbit polyclonal) | Proteintech | Proteintech: 16646–1-AP; RRID:AB_2756528 | WB (1:1000) |
| Antibody | anti-pAPN (rabbit polyclonal) | ABclonal | Abclonal: A5662; RRID:AB_2766422 | WB (1:500) |
| Antibody | anti-GAPDH (rabbit monoclonal) | Cell Signaling | Cell Signaling: 3683; RRID:AB_1642205 | WB (1:1000) |
| Antibody | anti-PRRSV-N (mouse monoclonal) | *Jiang et al., 2010* | | WB (1:2000) (Made in our laboratory) |
| Antibody | anti- PDCoV-N (mouse monoclonal) | *Luo et al., 2016* | | WB (1:1000) (Made in our laboratory) |
| Antibody | anti-β-actin (rabbit monoclonal) | ABclonal | Abclonal: AC026; RRID:AB_2768234 | WB (1:5000) |
| Antibody | anti-PDCoV-N (mouse monoclonal) | *Luo et al., 2016* | | IFA (1:100) (Made in our laboratory) |
| Antibody | anti-CD163 (mouse monoclonal) | Bio-Rad | Bio-Rad: MCA2311PE; RRID:AB_1510025 | Flow cytometry (1:10) |
| Antibody | anti-pAPN (rabbit monoclonal) | Abcam | Abcam: ab108310; RRID:AB_10866195 | IHC (1:300) |

*Continued on next page*

*Continued*

| Reagent type (species) or resource | Designation | Source or reference | Identifiers | Additional information |
|---|---|---|---|---|
| Antibody | anti-PRRSV-N (mouse monoclonal) | State Key Laboratory of Agricultural Microbiology, College of Veterinary Medicine, Huazhong Agricultural University, Wuhan, China | | IHC (1:800) (Made in our laboratory) |
| Recombinant DNA reagent | pX330 vector | Addgene | Addgene: #42230 | |
| Commercial assay or kit | enzyme-linked immunosorbent assay (ELISA) kit for PRRSV antibody detection | IDEXX Laboratories Inc | DEXX Laboratories: 99–40959 | |
| Commercial assay or kit | enzyme-linked immunosorbent assay (ELISA) kit for detection of the content of Hp in serum | Alpha Diagnostic | Alpha Diagnostic: 6250–40 | |
| Software, algorithm | GraphPad Prism software | GraphPad Prism | Version 6.0.0; RRID:SCR_002798 | |

## CRISPR constructs

For the *CD163* gene, the sgRNA was designed to target exon 7, and for the *pAPN* gene, the sgRNA was designed to target exon 2. The sequences of the two sgRNAs are as follows: GGAAACC-CAGGCTGGTTGGAGGG (*CD163*-sgRNA) and GCATCCTCCTCGGCGTGGCGG (*pAPN*-sgRNA). The PAM is indicated in bold font. The two sgRNA sequences were cloned into the pX330 vector (Addgene plasmid # 42230) and named pX330-*CD163* and pX330-*pAPN*, respectively. Two plasmids were extracted (TIANGEN, DP117) in large quantities and used to transfect the fetal fibroblasts of Large White pigs.

## Cell lines

LLC-PK1 cells and ST cells were obtained from American Type Culture Collection (LLC-PK1 cells: ATCC CL-101; ST cells: ATCC CRL-1746). Authentication of the cell lines was performed by STR profiling and had a negative mycoplasma contamination testing status.

## Cell transfection and selection

The fetuses of Large White pigs at 35-day-old were used to isolate PEFs, which were then cultured in DMEM medium containing 20% FBS. When the cells grew to 80% confluence, approximately $10^6$ cells were transfected with pX330-*CD163* (2.5 ug) and pX330-*pAPN* (2.5 ug) plasmids. A Lonza 2B nuclear transfection system was used for transfection with Nucleofector program T-016. The entire transfection process was performed according to the kit instructions (Lonza, VPI-1002). Cells were cultured for 48 hr after transfection and then seeded into 10 cm dishes at a density of 150 cells/dish. The culture medium was changed every 3 days, and cells were cultured for 10 days to form single-cell colonies. Single-cell colonies were transferred to 48-well plates for expansion culture. When cells in the 48-well plates reached confluence, 1/3 of the cells were taken for genotype identification, and the remaining cells continued to expand. Cells with genotypes identified as double-gene mutations were cultured and frozen for SCNT.

## Somatic cell nuclear transfer (SCNT)

The oocytes for SCNT were derived from a nearby slaughterhouse, and the nuclear donor cells were the DKO fibroblasts. The nuclear transfer donor cells were transferred into enucleated oocytes, and the reconstructed embryos were activated and cultured to develop into blastocysts. We then selected well-developed recombinant embryo clones to be surgically transferred into the oviduct of recipient gilts on the day after estrus was observed. After the embryo transfer, the technicians observed the estrus of the sow, and regularly checked the pregnancy by B-ultrasound.

## Genotyping

The *CD163* and *pAPN* genotypes of colonies and piglets born after nuclear transfer were detected by PCR and Sanger sequencing. One third of the cells in the 48-well plate and the ear tissue were used to extract the genomic DNA. The primer pairs *CD163*-F/*CD163*-R and *pAPN*-F/*pAPN*-R were used to amplify the sequences near the sgRNA target sites in the *CD163* and *pAPN* genes, respectively. The primer sequences are shown in *Supplementary file 5*. The PCR products were genotyped by Sanger sequencing.

## Off-target analysis

Potential off-target sites were predicted using an online software: CRISPOR (http://crispor.tefor.net/). We identified the 10 potential off-target sites for each of the two sgRNAs. Twenty pairs of primers were designed to amplify the potential off-target sites from the genomic DNA isolated from the 3 DKO pigs (1143#, 1144#, 1145#). Sanger sequencing was performed to determine whether any mutations occurred. The primer sequences are shown in *Supplementary file 7*.

## Western blotting

The total protein extracted from lung tissue, liver tissue, and spleen tissue of non-challenged WT pigs and DKO pigs was used to detect CD163, and protein extracted from duodenal, jejunal, and ileal tissues were used to detect pAPN expression. Whole cell lysates of PRRSV-infected PAMs and PDCoV-infected PAMs were used to quantify the expression levels of PRRSV nucleocapsid (N) protein and PDCoV nucleocapsid (N) protein, respectively. The protein samples were separated by 8% or 12% SDS-PAGE and transferred to a polyvinylidene fluoride membrane (Millipore). The membrane was blocked with 5% skim milk for 2 hr, and then incubated with primary antibody at 4°C overnight and secondary antibody at room temperature for 2 hr. Chemiluminescent signals were developed with SuperSignal West Pico PLUS Chemiluninescent Substrate (Thermos Scientific) and captured with a Tanon-520 (Tanon). CD163 rabbit polyclonal antibody (16646–1-AP; Proteintech) was used to detect porcine CD163. APN polyclonal antibody (A5662; ABclonal) was used to detect pAPN, anti-PRRSV-N antibody (made in our laboratory) was used to detect PRRSV-N protein, anti-PDCoV-N antibody (made in our laboratory) was used to detect PDCoV-N protein, GAPDH rabbit antibody (3683; Cell Signaling) or β-actin rabbit antibody (AC026; Abclonal Technology) was used to stain GAPDH or β-actin as a loading control. HRP-conjugated affinipure goat anti-rabbit IgG(H+L) (SA00001-2; Proteintech) and HRP-conjugated affinipure goat anti-mouse IgG(H+L) (SA00001-1; Proteintech) were used as the secondary antibody.

## Infection of PAMs

PAMs were isolated from DKO piglets and WT piglets. The lungs were obtained from the euthanized piglets. The lung surfaces were rinsed with PBS, and PAMs were subsequently obtained by bronchoalveolar lavage with PRMI-1640 medium (Gibco, USA). The collected lavage solution was dispensed into a 50 mL centrifuge tube, centrifuged at 300 g for 10 min, and the supernatant was discarded. PAMs were washed again with PRMI-1640 medium and then frozen in cryopreservation solution containing 90% FBS and 10% DMSO. For further in vitro infection experiments, PAMs were cultured in RPMI-1640 medium with 10% FBS and $1 \times$ antibiotic antimycotic (15240062; Invitrogen) at 37°C/5% $CO_2$, and then infected with a highly pathogenic PRRSV (HP-RRSV) strain WUH3 (GenBank accession number HM853673) (*Li et al., 2009*) at a dose of MOI = 0.1 and PDCoV strain CHN-HN-2014 (GenBank accession number KT336560) (*Dong et al., 2016*) at a dose of MOI = 10. The production of progeny PRRSV was evaluated through western blot, and qRT-PCR assays, and the production of progeny PDCoV was eveluted through IFA, $TCID_{50}$, qRT-PCR and western blot assays.

## Flow cytometry

PAMs were fixed in 4% formaldehyde for 15 min at room temperature. The cells were then blocked with 2% BSA overnight at 4°C and incubated with mouse anti-pig CD163 mAbs (MCA2311PE; Bio-Rad) at 37°C for 1 hr in the dark. After washing with PBS three times, PAMs were resuspended in PBS and immediately analyzed using a BD FACSVerse flow cytometer (BD Biosciences, CA) and FlowJo software (TreeStar, CA).

## Infection of pigs

All WT pigs used in the infection experiment were born from natural breeding, and they were matched by age and breed with the DKO pigs. The four DKO and six WT pigs used for PRRSV WUH3 viral challenge were both about 45 days old. Viral inoculation was conducted by nasal intubation drip (2 mL: $10^6$ TCID$_{50}$/mL) and intramuscular injection (2 mL: $10^6$ TCID$_{50}$/mL). During the 14 days of PRRSV challenge, piglet rectal temperature and clinical symptoms data (feeding, breathing, defecation, mental state) were collected every morning. At the same time, piglet survival rate was recorded, blood was collected, and the piglets were weighed regularly. If any pigs died during the course of the PRRSV challenge, pictures were immediately taken and samples were collected. All surviving pigs were slaughtered at 14 days post-infection (dpi) and lung tissue was examined for disease symptoms.

Four DKO pigs and six WT pigs were used for TGEV challenge. Pigs were inoculated with a total of 10 mL of TGEV strain WH-1 (GenBank accession number HQ462571) (*An et al., 2014*) ($7 \times 10^5$ TCID$_{50}$/mL) that were orally administered to each pig in two doses (day 0 and day 1, 5 mL/day). For the PDCoV challenge, two DKO pigs and four WT pigs were orally administered a total of 16 mL of PDCoV strain CHN-HN-2014 ($2.5 \times 10^8$ TCID$_{50}$/mL) divided into two doses delivered on day 0 and day 1 (8 mL/day). For both the TGEV and PDCoV groups, the rectal temperature of the pigs was measured daily for the full 14 day experiment and the diarrhea of the piglets was observed. Blood was collected and the piglets were weighed regularly. In the TGEV group, a DKO pig and a WT pig were slaughtered on day 3, and the remaining pigs in the TGEV group and all pigs in the PDCoV group were slaughtered at 14 dpi. After slaughter, the pigs were dissected to observe the gross lesions in small intestine tissue, to collect small intestine tissue samples, and to detect any pathological changes by H and E staining. Meanwhile, during these 14 days, 4 WT pigs and 2 DKO pigs were reared under the same conditions without any virus infection, and these pigs were used as the Mock group.

## Hematoxylin and eosin (H and E) staining and immunohistochemistry (IHC)

Lung tissues of pigs in the PRRSV challenge group, and duodenum, jejunum, and ileum tissues in the TGEV and PDCoV groups were collected. The tissues were fixed in 4% paraformaldehyde fixative, dehydrated, embedded, and cut into 3 ~ 8 μm-thick sections. For histopathology, the sections were stained by H and E. For IHC, tissue sections were stained with antibodies specific to the corresponding protein antigens. Tissue sections were then observed and photographed with a fluorescence microscope. The antibodies used to detect pAPN protein were purchased from Abcam (ab108310); the antibody used to detect PRRSV-N protein was made by our laboratory.

## Measurement of viral antibody

The blood tissues of three experimental groups of pigs were collected at different times after viral challenge and the sera were separated. For the PRRSV group, the sera from all samples were subjected to PRRS antibody detection by commercially available enzyme-linked immunosorbent assay (ELISA) kit (IDEXX, ME). The antibody level was determined to be negative or positive according to the S/P value. If S/P<0.4, the antibody is negative, and if S/P≥0.4, the antibody is positive. In order to detect TGEV-specific and PDCoV-specific antibody levels in serum, we used a serum neutralization test (SNT). Briefly, sera were heat inactivated by 30 min of incubation in a 56°C water bath. Then serial 2-fold dilutions of serum samples in four replicates were mixed with 200 TCID$_{50}$ of TGEV strain WH-1 in a 1:1 ration. After incubation, 100 μl of the mixture was added into ST cells (a swine testicular cell line permissive of TGEV infection; ATCC CRL-1746) at a confluence of ~90%, seeded in 96-well cell culture plates. Appropriate serum, virus (200 TCID$_{50}$, 20 TCID$_{50}$, 2 TCID$_{50}$, and 0.2 TCID$_{50}$), and cell controls were included in this test. For about 72 hr after incubation, the cells were monitored for TGEV-specific cytopathic effects. Neutralization titers were calculated as the reciprocal of the highest dilution resulting in complete neutralization. Similarly, sera were diluted mixed with 200 TCID$_{50}$ of PDCoV strain CHN-HN-2014. In contrast, PDCoV titers were assessed using LLC-PK1 cells (a porcine kidney cell line permissive of PDCoV infection; ATCC CL-101) that were washed twice with Dulbecco's Modified Eagle's Medium (DMEM) (Invitrogen, CA), and supplemented with 7.5 μg/mL trypsin (Gibco, USA) prior to and after 1 hr incubation with these mixtures. Cells were then

cultured in DMEM supplemented with 7.5 µg/mL trypsin for approximately 72 hr, and the neutralization titers of sera from PDCoV group were calculated.

## qRT-PCR-based measurement of PRRSV RNA and PDCoV RNA

To quantify the copies of PRRSV and PDCoV in the infected experimental group, we extracted PRRSV RNA from PAMs, serum, lung tissue, and tonsil tissue from both the challenge and the mock-inoculated group, and extracted PDCoV RNA from DKO PAMs and WT PAMs after infected with PDCoV. RNA extraction was performed using TRIzol reagent (Omega Bio-Tek). The RNA was reverse transcribed into cDNA according to the instructions for a Transcriptor First Strand cDNA Synthesis Kit (Roche). The cDNA was then amplified with SYBR green real-time PCR master mix (Applied Biosystems) in an ABI 7500 real-time PCR system (Applied Biosystems). RNA copy numbers were calculated from a standard curve drawn from positive standards at different dilutions. The primers used for qRT-PCR are listed as follows: 5'-GCAATTGTGTCTGTCGTC-3' and 5'-CTTATCCTCCCTGAATCTGAC-3' for PRRSV; 5'-GCCCTCGGTGGTTCTATCTT-3' and 5'-TCCTTAGCTTGCCCCAAATA-3' for PDCoV.

## IFA assay

DKO PAMs and WT PAMs in 24-well cell culture plates were infected or mock-infected with PDCoV at a multiplicity of infection (MOI) of 10. At 24 hpi, cells were fixed with 4% paraformaldehyde for 15 min and permeabilized with methanol for 10 min at room temperature. The cells were then blocked with bovine serum albumin (5%) diluted in phosphate-buffered saline (PBS) for 1 hr, and incubated with a PDCoV-N-protein-specific monoclonal antibody for 1 hr and an Alexa Fluor 488-conjugated donkey anti-mouse IgG for 1 hr. The cell nuclei were counterstained with 4',6-diamidino-2-phenylindole (DAPI) for 15 min at room temperature. After three washes with PBS, the stained cells were observed with an inverted fluorescence microscope (Olympus IX73, Japan).

## TCID$_{50}$ assay

PDCoV-infected PAMs were frozen and thawed repeatedly to completely release viruses. Next, LLC-PK1 cells (a pig kidney cell line known to be highly permissive to PDCoV infection) were seeded in 96-well plates and were infected with 10-fold serial dilutions of virus samples in eight replicates. At 72 hpi, PDCoV titers were calculated based on cytopathic effects and expressed as the TCID$_{50}$ value per milliliter, using the Reed–Muench method.

## Measurement of haptoglobin (Hp)

The amount of Hp in serum was measured using an enzyme-linked immunosorbent assay (ELISA) kit (6250–40, Alpha Diagnostic) specific to pig Hp, as previously described (*Yang et al., 2018*). Assays were performed in triplicate for each sample.

## Carcass trait measurements and analysis of pork nutrition

The quality and performance of pigs related to slaughter were determined by a third-party testing center (The national breeding swine quality supervision and testing center (Chongqing), Ministry of Agriculture and Rural Affairs of China). All testing followed the guidelines stipulated in the 'rules for performance testing of breeding pigs' document published by the Ministry of Agriculture and Rural Affairs of PR China (NY/T 822–2004). Briefly, DKO pigs and control WT pigs were weighed before slaughter, euthanized after fasting for 24 hr, and hairs, heads, hoofs, and internal organs were removed after carcass dissection. The weight of carcass, length of carcass, loin eye areas, thickness of skin, and backfat thickness of carcass were all measured. Ham, skin, bone, lean, and fat were dissected from the left side of the carcass and their individual weights were determined. To evaluate meat quality, we measured muscle pH, meat color score, intramuscular fat, marbling, and drip loss of longissimus dorsi. For analysis of pork nutrition, total protein, total fat, ash, moisture, amino acid, and individual minerals, amino acids were analyzed for the longissimus dorsi. The nutritional content of the pork was tested by the Beijing Institute of Nutritional Sources.

### Detection of sperm motility

Semen from DKO pigs and WT control pigs were collected and returned to the laboratory in a 17°C incubator for testing their quality. The detection system was Hamilton-Thorne Research IVOS II computer-assisted sperm analyzer to measure the concentration, motility, and velocity distribution of the sperm.

### Statistical analysis

All data are presented as the mean ± standard error of mean (SEM). Data from each of the two groups of pigs were compared with an unpaired t-test when a normal distribution was not obtained. The significance levels were set at 0.05, 0.01, and 0.001, as indicated by *, **, ***, respectively. The data was analyzed with GraphPad Prism 6.0.0 for Windows (GraphPad Software, La Jolla, California).

## Acknowledgements

This work was financially supported by the National Transgenic Breeding Project (2016ZX08010–004), the Major Scientific Research Tasks for Scientific and Technological Innovation Projects of the Chinese Academy of Agricultural Sciences (CAAS-ZDRW202006), Shandong Landsee Genetics Co., Ltd. (Kh17134), the National Natural Science Foundation of China (U1704231), the Shenzhen Key Technology Projects (JSGG20180507182028625), the National Transgenic Breeding Project (2016ZX08006–001), and the National Natural Science Foundation of China ( 31490602).

## Additional information

### Competing interests

Changli Ge: is affiliated with Shandong Landsee Genetics Co., Ltd. The author has no financial interests to declare. Haitao Shang: is affiliated with Shenzhen Kingsino Technology Co., Ltd. The author has no financial interests to declare. The other authors declare that no competing interests exist.

### Funding

| Funder | Grant reference number | Author |
| --- | --- | --- |
| National Transgenic Breeding Project | 2016ZX08010-004 | Yulian Mu |
| Chinese Academy of Agricultural Sciences | Major Scientific Research Tasks for Scientific and Technological Innovation Project CAAS-ZDRW202006 | Yulian Mu |
| Shandong Landsee Genetics Co., Ltd. | Kh17134 | Kui Li |
| National Natural Science Foundation of China | U1704231 | Liurong Fang |
| Shenzhen Key Technology Projects | JSGG20180507182028625 | Haitao Shang |
| National Transgenic Breeding Project | 2016ZX08006-001 | Kui Li |
| National Natural Science Foundation of China | 31490602 | Shaobo Xiao |

The funders had no role in study design, data collection and interpretation, or the decision to submit the work for publication.

### Author contributions

Kui Xu, Resources, Data curation, Formal analysis, Methodology, Writing - original draft, Project administration; Yanrong Zhou, Data curation, Formal analysis, Methodology, Writing - original draft, Writing - review and editing; Yulian Mu, Conceptualization, Resources, Data curation, Formal

analysis, Investigation, Project administration, Writing - review and editing; Zhiguo Liu, Shaohua Hou, Resources, Formal analysis, Investigation; Yujian Xiong, Yinghui Wei, Xiuling Zhang, Changjiang Xu, Jingjing Che, Ziyao Fan, Guangming Xiang, Jiankang Guo, Resources, Data curation, Investigation; Liurong Fang, Conceptualization, Resources, Supervision, Methodology; Changli Ge, Resources, Project administration; Haitao Shang, Resources, Data curation; Hua Li, Resources, Formal analysis, Validation, Methodology; Shaobo Xiao, Conceptualization, Resources, Data curation, Formal analysis, Supervision, Methodology, Project administration, Writing - review and editing; Julang Li, Conceptualization, Formal analysis, Validation, Methodology, Writing - review and editing; Kui Li, Conceptualization, Resources, Funding acquisition, Project administration, Writing - review and editing

### Author ORCIDs

Kui Xu https://orcid.org/0000-0003-2804-6126
Zhiguo Liu http://orcid.org/0000-0003-0599-9039
Haitao Shang https://orcid.org/0000-0001-5532-8335
Kui Li https://orcid.org/0000-0002-5686-3898

### Ethics

Animal experimentation: All experimental protocols related to animal work described in this study were reviewed and approved by the Institutional Animal Care and Use Committee (IACUC) at the Institute of Animal Sciences, Chinese Academy of Agricultural Sciences. All experiments were performed in accordance with the approved guidelines for animal care and management of research projects. (IAS2018-12 ).

### Decision letter and Author response

Decision letter https://doi.org/10.7554/eLife.57132.sa1
Author response https://doi.org/10.7554/eLife.57132.sa2

## Additional files

### Supplementary files

• Source data 1. Amino acid content of DKO lean meat and WT lean meat.

• Source data 2. Birth weights and average daily gains of WT pigs and DKO pigs from birth weight to slaughtering weight.

• Supplementary file 1. Genotypes of cell lines used in somatic cell nuclear transfer.

• Supplementary file 2. Embryo transfer data for *CD163* and *pAPN* DKO pigs.

• Supplementary file 3. Potential off-target sites identified for *CD163* sgRNA and *pAPN* sgRNA.

• Supplementary file 4. Amino acid content of DKO lean meat and WT lean meat.

• Supplementary file 5. Primers for genotyping.

• Supplementary file 6. Primers for PCR detection of random integration.

• Supplementary file 7. Primers for off-target analysis.

• Supplementary file 8. Birth weights and average daily gains of of WT pigs and DKO pigs from birth weight to slaughtering weight.

• Transparent reporting form

### Data availability

All data generated or analysed during this study are included in the manuscript and supporting files. Source data files have been provided for Figure 2, Figure 3, Figure 4, Figure 4-figure supplement 1, Figure 5, Table 1, Table 2 and Table 3.

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
