## [Decision Letter]

**Acceptance summary:**

In this study, the authors generated double gene knockout (DKO) pigs ablating expression of CD163 and pAPN and show complete resistance to PRRSV and TGEV, which has with clear applications in agriculture. Importantly, the authors also show partial resistance to challenge by pDCoV providing functional, in vivo data highlighting the likely role of pAPN as cross-genus coronavirus receptor. The manuscript utilizes promising CRISPR technology in livestock and provides novel information on coronavirus receptor usage in vivo.

**Decision letter after peer review:**

Thank you for submitting your article "*CD163* and *pAPN* double-gene-knockout pigs are resistant to three viruses while maintaining normal production performance" for consideration by *eLife*. Your article has been reviewed by three peer reviewers, one of whom is a member of our Board of Reviewing Editors, and the evaluation has been overseen by Karla Kirkegaard as the Senior Editor. The following individuals involved in review of your submission have agreed to reveal their identity: Christopher K Tuggle (Reviewer #2).

The reviewers have discussed the reviews with one another, and the Reviewing Editor has drafted this decision to help you prepare a revised submission.

Summary:

In this study, the authors generated double gene knockout (DKO) pigs ablating expression of CD163 and pAPN. These proteins are known receptors for the pig coronaviruses PRRSV and TGEV, respectively. Whether pAPN also serves as receptor for pDCoV in vivo was unknown. This is an important area of research, as pig coronaviruses cause major economic losses to pig production and the potential of coronaviruses to cross species and cause human disease is significant. Using the DKO pigs, the authors show CD163 and pAPN ablation results in strong resistance to PRRSV and TGEV infection as characterized by pathology, viral load determination and the generation of neutralizing antibodies. Importantly, the authors show that DKO pigs are partially resistant to challenge by pDCoV providing functional, in vivo data highlighting the likely role of pAPN as cross-genus coronavirus receptor. Overall, this manuscript is clear, well-written and the methods are robust. It utilizes promising CRISPR technology in life stock and provides novel information on coronavirus receptor usage in vivo. Furthermore, it demonstrates complete resistance of two major pig viruses with clear applications in agriculture. There are few areas that need to be clarified.

Essential revisions:

1) The phenotype of pDCoV infection in the DKO pigs is the first demonstration that pAPN could also be important for in vivo pDCoV infection. However, the resistance was only partial since antibody titers showed a delay but were similar at later time points and no significant differences were found in pathology. Effects on viral titer or weight loss were not reported. One can even argue if this merits the term "partial resistance" and whether "decreased susceptibility" would be more appropriate. Therefore, the authors should soften their claims throughout the manuscript. Importantly the title must be changed because in its current form it is incorrectly suggests that the resistance to pDCoV is of the same magnitude as the resistance to PRRSV and TGEV.

2) The potential in vivo role of pAPN was only assessed in the double knockout context together with CD163 knockout. The authors cannot rule out an effect of CD163 in the delayed adaptive immune response observed. The authors should explicitly state this and discuss the potential of CD163 KO to perturb an immune function, as an alternative explanation for the delayed immune response.

3) The authors make statements that go beyond what they have presented. The authors state several times in Results section and Discussion section that growth was not affected in the DKO pigs, yet no data is presented on this. For example: "Taken together, with the exception of slight meat coloring score increase, these results show that the simultaneous, editing-based disruption of the CD163 and pAPN loci, does not affect the normal growth and reproductive performance of the resultant DKO pigs" At minimum, the age of the pigs reported in Table 1 where weights are given. It would be better to actually document growth rate up to market weight; they seem to have adult male pigs given the semen evaluation. In this context, it was further stated: "we observed no difference in the production performance, reproductive performance, or pork nutrient contents between DKO pigs and WT pigs." However, iron is a nutrient, and this statement should be adjusted since they found substantial differences in iron content.

4) Infection of pigs. "The four DKO and six WT pigs used for PRRSV WUH3 viral challenge were both about 45 days old. "What was the source of the WT pigs? Were they also SCNT-derived? If yes, this should be indicated. If not, there are two differences between the groups of pigs, not just genotype. This should be acknowledged.

Revisions expected in follow-up work:

1) The partial resistance to PDCoV is intriguing in the DKO pigs but only found in the delayed adaptive immune response. In follow up work, it would be important to distinguish better the effects of single pAPN knockout versus DKO pigs in vivo. In addition, isolating primary cells from the WT and DKO pigs and do an in vitro challenge with PDCoV would quantify more directly the impact of DKO on virus infection.

---

## [Author Response]

Summary:In this study, the authors generated double gene knockout (DKO) pigs ablating expression of CD163 and pAPN. These proteins are known receptors for the pig coronaviruses PRRSV and TGEV, respectively. Whether pAPN also serves as receptor for pDCoV in vivo was unknown. This is an important area of research, as pig coronaviruses cause major economic losses to pig production and the potential of coronaviruses to cross species and cause human disease is significant. Using the DKO pigs, the authors show CD163 and pAPN ablation results in strong resistance to PRRSV and TGEV infection as characterized by pathology, viral load determination and the generation of neutralizing antibodies. Importantly, the authors show that DKO pigs are partially resistant to challenge by pDCoV providing functional, in vivo data highlighting the likely role of pAPN as cross-genus coronavirus receptor. Overall, this manuscript is clear, well-written and the methods are robust. It utilizes promising CRISPR technology in life stock and provides novel information on coronavirus receptor usage in vivo. Furthermore, it demonstrates complete resistance of two major pig viruses with clear applications in agriculture. There are few areas that need to be clarified.Essential revisions:1) The phenotype of pDCoV infection in the DKO pigs is the first demonstration that pAPN could also be important for in vivo pDCoV infection. However, the resistance was only partial since antibody titers showed a delay but were similar at later time points and no significant differences were found in pathology. Effects on viral titer or weight loss were not reported. One can even argue if this merits the term "partial resistance" and whether "decreased susceptibility" would be more appropriate. Therefore, the authors should soften their claims throughout the manuscript. Importantly the title must be changed because in its current form it is incorrectly suggests that the resistance to pDCoV is of the same magnitude as the resistance to PRRSV and TGEV.

First, we would like to thank the reviewers for support of our work and the constructive suggestions. Regarding this first point, we have now softened the claims about the susceptibility of DKO pigs to PDCoV and have replaced the term "partial resistance" with "decreased susceptibility" throughout the manuscript. We have changed the title "*CD163* and *pAPN* double-gene-knockout pigs are resistant to three viruses while maintaining normal production performance" to "*CD163* and *pAPN* double-knockout pigs are resistant to PRRSV and TGEV and exhibit decreased susceptibility to PDCoV while maintaining normal production performance".

2) The potential in vivo role of pAPN was only assessed in the double knockout context together with CD163 knockout. The authors cannot rule out an effect of CD163 in the delayed adaptive immune response observed. The authors should explicitly state this and discuss the potential of CD163 KO to perturb an immune function, as an alternative explanation for the delayed immune response.

Thanks for focusing our attention on this very important point. Indeed, despite the important role of CD163 in innate immunity, an inhibiting effect of soluble CD163 on the adaptive immune system has also been reported (Frings et al., 2002; O'Connell et al., 2017). It is possible that the delayed adaptive immune response that we observed in PDCoV-infected DKO pigs may be associated with *CD163* knockout-induced immunosuppression. We have added this alternative explanation to the Discussion section of the revised manuscript, for example adding: "…indicating the potential role of *pAPN* in mediating PDCoV infection. Additionally, the effect of *CD163* knockout in the delayed adaptive immune response cannot be ignored. Despite the important role of CD163 in innate immunity, an inhibiting effect of soluble CD163 on the adaptive immune system has also been reported (Frings et al., 2002; O'Connell et al., 2017). It is thus possible that the delayed adaptive immune response we observed in PDCoV-infected DKO pigs may be associated with *CD163* knockout-induced immunosuppression."

Please also note that, seeking to further confirm the potential mechanisms involved in this delayed adaptive immune response, we have recently initiated a project to generate *pAPN* KO pigs as suggested by reviewers; we plan to use these animals to further elucidate the role of pAPN in PDCoV infection. As guided by the *eLife* editor notes, we plan to report this result when it is ready, either as a preprint on bioRxiv or potentially as a Research Advance in *eLife*.

3) The authors make statements that go beyond what they have presented. The authors state several times in Results section and Discussion section that growth was not affected in the DKO pigs, yet no data is presented on this. For example: "Taken together, with the exception of slight meat coloring score increase, these results show that the simultaneous, editing-based disruption of the CD163 and pAPN loci, does not affect the normal growth and reproductive performance of the resultant DKO pigs" At minimum, the age of the pigs reported in Table 1 where weights are given. It would be better to actually document growth rate up to market weight; they seem to have adult male pigs given the semen evaluation. In this context, it was further stated: "we observed no difference in the production performance, reproductive performance, or pork nutrient contents between DKO pigs and WT pigs." However, iron is a nutrient, and this statement should be adjusted since they found substantial differences in iron content.

Thanks for the guidance in clarifying these important points. According to this guidance, we have added the age of the pigs in Table 1; we have also added a Supplementary file 8 in which we list the data for the birth weights and average daily gains of WT pigs and DKO pigs up to their slaughtering weight. Note that inferential statistical analysis revealed no significant differences between the two groups for these growth parameters. We have also revised the manuscript, which now states

"In addition, there was no significant difference in birth weight or in the average daily gain between WT and DKO pigs (Supplementary file 8)."

And

"With the exception of meat color score and iron content, no differences in the production performance, reproductive performance, or pork nutrient content were observed between DKO pigs and WT pigs."

4) Infection of pigs. "The four DKO and six WT pigs used for PRRSV WUH3 viral challenge were both about 45 days old. "What was the source of the WT pigs? Were they also SCNT-derived? If yes, this should be indicated. If not, there are two differences between the groups of pigs, not just genotype. This should be acknowledged.

We now understand the lack of clarity for this content in our originally submitted text. To clarify, the WT pigs used in the infection experiment were from natural breeding; they were matched by age and breed with the DKO pigs. We have added the source of the WT pigs to the beginning of the "Infection of pigs" part of Materials and methods section in the revised manuscript. The new sentence reads: "All WT pigs used in the infection experiment were born from natural breeding, and they were matched by age and breed with the DKO pigs."

Revisions expected in follow-up work:1) The partial resistance to PDCoV is intriguing in the DKO pigs but only found in the delayed adaptive immune response. In follow up work, it would be important to distinguish better the effects of single pAPN knockout versus DKO pigs in vivo. In addition, isolating primary cells from the WT and DKO pigs and do an in vitro challenge with PDCoV would quantify more directly the impact of DKO on virus infection.

Thanks for this excellent suggestion. As guided by the suggestion about "isolating primary cells from the WT and DKO pigs and do an in vitro challenge with PDCoV would quantify more directly the impact of DKO on virus infection", we have now completed this experimental work and have added a description of our new findings in the revised Results and Discussion sections. The added sentences in the Results section are as follows:

"In addition, we tested the resistance of PAMs derived from DKO pigs to PDCoV. DKO and WT PAMs were infected with PDCoV, and indirect immunofluorescence assays (IFA), tissue culture infectious dose 50 (TCID_50_) assays, qRT-PCR, and western blot analyses to assess PDCoV proliferation in PAMs all indicated that DKO PAMs exhibit significantly decreased susceptibility of PDCoV infection compared to WT PAMs (Figure 4—figure supplement 1)".

The revised sentences in the Discussion section are as follows:

"Interestingly, a recent study showed that PAMs, but not lung fibroblast-like cells, from pAPN knockout pigs showed resistance to PDCoV infection (Stoian et al., 2020), a finding consistent with our in vitro experiments showing that DKO PAMs exhibit decreased susceptibility to PDCoV infection".

The figure content for these new in vitro experiments (now added to the revised manuscript) includes the following figure caption:

" Figure 4—figure supplement 1 | PAMs of DKO pigs exhibit reduced susceptibility to PDCoV. (A) WT PAMs and DKO PAMs were infected or mock-infected with PDCoV (MOI = 10); at 24 hpi, PDCoV-N-specific fluorescence signals were detected by IFA. (B) WT PAMs and DKO PAMs were infected with PDCoV (MOI = 10); at 24 hpi, cells were collected and the viral titer was determined by TCID_50_ assays (in LLC-PK1 cells). (C, D) WT PAMs and DKO PAMs were infected with PDCoV (MOI = 10); at 24 hpi, cells were harvested and analyzed using qRT-PCR (C) and western blot assays (D). The PDCoV-N protein level was quantified by ImageJ. Data are expressed as the mean ± SD. Statistical significance was determined by Student’s *t* test; ns, *P* > 0.05; **P* < 0.05; ***P* < 0.01; ****P* < 0.001."

We have also added the corresponding original data in the source data files for the revised manuscript.

We have also made some minor modifications in the Materials and methods section including "Western blotting", "Infection of PAMs", and "qRT-PCR-based measurement of PRRSV RNA and PDCoV RNA" parts using tracked changes. And we have added experimental methods including "IFA assay" and "TCID_50_ assay". The new methods content reads as follows:

"DKO PAMs and WT PAMs in 24-well cell culture plates were infected or mock-infected with PDCoV at a multiplicity of infection (MOI) of 10. At 24 hpi, cells were fixed with 4% paraformaldehyde for 15 minutes and permeabilized with methanol for 10 min at room temperature. The cells were then blocked with bovine serum albumin (5%) diluted in phosphate-buffered saline (PBS) for 1 hour, and incubated with a PDCoV-N-protein-specific monoclonal antibody for 1 hour and an Alexa Fluor 488-conjugated donkey anti-mouse IgG for 1 hour. The cell nuclei were counterstained with 4’,6-diamidino-2-phenylindole (DAPI) for 15 minutes at room temperature. After three washes with PBS, the stained cells were observed with an inverted ﬂuorescence microscope (Olympus IX73, Japan)."

And

"PDCoV-infected PAMs were frozen and thawed repeatedly to completely release viruses. Next, LLC-PK1 cells (a pig kidney cell line known to be highly permissive to PDCoV infection) were seeded in 96-well plates and were infected with 10-fold serial dilutions of virus samples in eight replicates. At 72 hpi, PDCoV titers were calculated based on cytopathic effects and expressed as the TCID_50_ value per milliliter, using the Reed–Muench method."

Please also note that we have now initiated work to generate *pAPN* single knockout pigs to exclude the effects of *CD163* knockout. We have obtained *pAPN* knockout pig fetal fibroblasts (PEFs); however, owing to the ongoing epidemics of African swine fever and COVID-19, there has been delay in our progress. The work is ongoing, and as guided by the *eLife* editor notes, we plan to report this result when it is ready, either as a preprint on bioRxiv or potentially as a Research Advance in *eLife*.